# Mo_2_TiAlC_2_ as a Saturable Absorber for a Passively Q-Switched Tm:YAlO_3_ Laser

**DOI:** 10.3390/nano14221823

**Published:** 2024-11-14

**Authors:** Chen Wang, Tianjie Chen, Zhe Meng, Sujian Niu, Zhaoxue Li, Xining Yang

**Affiliations:** Xinjiang Key Laboratory for Luminescence Minerals and Optical Functional Materials, School of Physics and Electronic Engineering, Xinjiang Normal University, Urumqi 830054, China; wangchen0005@163.com (C.W.); chentianjie0131@sina.com (T.C.); meng13688626273@163.com (Z.M.); nsj@xjnu.edu.cn (S.N.)

**Keywords:** MAX phase, saturable absorber, passively Q-switched, pulsed laser

## Abstract

Owing to their remarkable characteristics, two-dimensional (2D) layered, MAX phase materials have garnered significant attention in the field of optoelectronics in recent years. Herein, a novel MAX phase ceramic material (Mo_2_TiAlC_2_) was prepared into a saturable absorber (SA) by the spin-coating method for passively Q-switching (PQS), and its nonlinear optical absorption properties were characterized with a Tm:YAlO_3_ (Tm:YAP) nanosecond laser. The structure characteristics and composition analysis revealed that the Mo_2_TiAlC_2_ material exhibits a well-defined and stable structure, with a uniform thin film successfully obtained through spin coating. In this study of a PQS laser by employing a Mo_2_TiAlC_2_-based SA, an average output power of 292 mW was achieved when the absorbed pump power was approximately 4.59 W, corresponding to a central output wavelength of 1931.2 nm. Meanwhile, a stable pulse with a duration down to 242.9 ns was observed at a repetition frequency of 47.07 kHz, which is the narrowest pulse width recorded among PQS solid-state lasers using MAX phase materials as SAs. Our findings indicate that the Mo_2_TiAlC_2_ MAX phase ceramic material is an excellent modulator and has promising potential for ultrafast nonlinear photonic applications.

## 1. Introduction

Two-dimensional (2D) materials, such as semiconductors, transition metal dichalcogenides, and topological insulators, exhibit distinct properties compared to traditional materials, including a high specific surface area, unique electronic band structures, flexibility, and ultra-lightweight [1,2,3,4,5]. In recent years, they have shown considerable promise and significant potential for applications in sensing, catalysis, biomedical, and energy storage [6,7,8,9]. The MAX phases, as an attractive type of 2D ceramic materials, are a large family of hexagonal, ternary, carbides, and nitrides with a general formula of M*_n_*_+1_AX*_n_* (MAX), where *n* = 1, 2, 3, etc.; M stands for the transition metal (Sc, Ti, Zr, Hf, V, Nb, Ta, Cr, Mo, etc.); A represents a group Ⅲ, Ⅳ, Ⅴ, or Ⅵ element (Al, Ga, In, Tl, Si, Ge, Sn, Pb, P, As, etc.); and X is carbon or nitrogen or a combination thereof [10,11,12]. These phases display both exceptional metal and ceramic properties, including high thermal and electrical conductivity, outstanding oxidation resistance, and remarkable damage tolerance [13,14,15,16]. Furthermore, the localized electronic characteristics of MAX phase materials, such as their ultrafast carrier relaxation processes, exciton effects, and tunable optical responses, render them valuable for applications in optics and photonics, as all-optical switches, light-emitting diodes, and SAs [17,18,19,20].

For instance, various MAX phase-based SAs, such as Ti_2_AlC, Ti_2_AlN, Cr_2_TiAlC_2_, and Ta_2_AlC, have been extensively investigated as modulators in PQS all-fiber lasers operating in the approximately 1.5 μm wavelength region, with the corresponding pulse widths typically spanning from microseconds to seconds [21,22,23,24]. Following that, PQS all-fiber lasers utilizing V_4_AlC_3_ and Ti_3_AlC_2_ as SAs have been extended to the mid-infrared 2 μm waveband, thereby encompassing a broader range of applications, including material processing, laser surgery, and remote sensing [25,26,27,28,29]. To date, a pulse width as narrow as 770 ns [23] and an average output power as high as 750 mW [25] have been achieved by employing various MAX phases as the SAs in PQS all-fiber lasers. In another report published in 2023, a passively mode-locked (PML) fiber laser was studied by using Cr_2_AlC as the SA, and an ultrashort pulse with a duration down to 1.3 ps was obtained with a central output wavelength of 1.9 μm [30]. These studies have illustrated that outstanding output performances of PQS and PML fiber lasers can be achieved by utilizing varieties of 2D MAX phase ceramic materials as SAs. However, few studies of pulsed solid-state lasers with MAX phases as SAs have been reported, and the underlying mechanism has yet to be fully explored. Mo_2_TiAlC_2_, a novel addition to the MAX phase family, has recently attracted considerable attention owing to its ultrafast carrier dynamics and broad nonlinear optical response that spans from the visible to mid-infrared regions [31,32,33]. The exploration of Mo_2_TiAlC_2_-SA as a modulator for a PQS solid-state laser remains unreported until now, despite extensive investigations into the nonlinear response of this substance [34,35].

In this study, a facile spin-coating method was employed to fabricate Mo_2_TiAlC_2_-SA, and its application in a PQS Tm:YAP laser was investigated. Mo_2_TiAlC_2_-SA was specially selected due to its broader absorption spectrum, enhanced modulation depth, and potential applications in mid-infrared pulsed lasers. Our results indicate that MAX phase materials exhibit superior saturation absorption responses in PQS solid-state lasers compared to all-fiber lasers [36]. In the PQS mode, a stable ultrashort pulse with a duration of 242.9 ns was achieved, marking the shortest pulse recorded among PQS lasers utilizing 2D MAX phase ceramic materials as SAs, and an average output power of 292 mW was obtained at 1931.2 nm. Furthermore, we propose that this research paves new avenues for the applications of MAX phase ceramic materials in pulsed solid-state lasers and highlights their potential as modulators with exceptional nonlinear optical properties in the field of optoelectronics.

## 2. Experimental Section

### 2.1. Preparation of Mo_2_TiAlC_2_-Based SA

The Mo_2_TiAlC_2_ MAX phase material, supplied by Xianfeng Nanotechnology Co., Ltd. (Nanjing, China), is composed of well-ordered trivalent carbide atomic layers with a composition ratio of Mo:Ti:Al:C = 6:1.5:1:2, and particle sizes ranging from 1 to 5 µm. In Mo_2_TiAlC_2_, each unit cell contains 12 atoms. The M and A atomic layers are arranged in an alternating pattern to form a layered structure akin to a close-packed hexagonal structure, with the X atom occupying the interstitial position of the octahedron. The synthesis of the Mo_2_TiAlC_2_ MAX phase was primarily achieved through ball milling of the raw material powder followed by high-temperature sintering. Furthermore, for the preparation of the Mo_2_TiAlC_2_-based SA, anhydrous ethanol with a purity greater than 99.7% was used as the solvent, sourced from Beilian Fine Chemicals Development Co., Ltd. (Tianjin, China). The preparation process for the Mo_2_TiAlC_2_-based SA is depicted in Figure 1. Initially, a 20 mg sample of Mo_2_TiAlC_2_ powder with a purity of 98 wt% was precisely weighed and combined with 20 mL of anhydrous ethanol in a vial. The resulting mixture underwent ultrasonic treatment for 16 h using an ultrasonic cell pulverizer (LICHEN, JY96-IIN, Shanghai, China). Subsequently, the solution obtained after ultrasonication was allowed to settle for 24 h to isolate the supernatant. This supernatant was then applied to a CaF_2_ substrate utilizing a spin coater (SETCAS, KW-4A, Beijing, China), involving four cycles at low speed (30 rpm for 10 s) followed by high speed (100 rpm for 10 s). After drying at room temperature for a while, a high-quality SA based on Mo_2_TiAlC_2_ was successfully prepared.

### 2.2. Characterization of Mo_2_TiAlC_2_ Material

To investigate the basic microstructure of the Mo_2_TiAlC_2_ powder, a scanning electron microscopy (SEM, ZEISS, Sigma 300, Oberkochen, BW, Germany) was employed for the high-resolution characterization of its surface morphology and particle structure. As shown in Figure 2a–c, the Mo_2_TiAlC_2_ MAX phase nanosheets are distinctly observable, displaying micron-scale particle sizes and a relatively uniform distribution. The multi-layered accordion-like architecture is evident at higher magnifications, where the characteristic 2D lamellar structure becomes clearly discernible. Figure 2d,e demonstrate the application of energy dispersion spectroscopy (EDS, ZEISS, Sigma 300, Oberkochen, BW, Germany) to ascertain the elemental composition of the Mo_2_TiAlC_2_ through point and line scan analyses, respectively. A previous analysis showed that molybdenum constituted 57.63 wt.% of the composition, followed by titanium at 14.97 wt.%, aluminum at 9.49 wt.%, and carbon at 17.92 wt.%. No significant impurities were detected, suggesting a commendable level of material purity. A subsequent analysis revealed the concentration curves for each element along the scan line, further confirming particle uniformity as evidenced by the SEM.

A detailed microstructure analysis of this material is illustrated in Figure 3, utilizing transmission electron microscopy (TEM, JEOL, JEM-F200, Tokyo, HI, Japan). The Mo_2_TiAlC_2_ nanosheets exhibit remarkable transparency when subjected to electron beam irradiation, and are noted to have fewer layers. These layers exhibit interstitial spacing with evident folds at the edge, and the transverse particle size remains uniform, as observed in the SEM. Moreover, the high-resolution images reveal the distinct rough surface features and a fingerprint-like texture on the individual Mo_2_TiAlC_2_ particles, while the low-resolution images depict the MAX phase sheets adhering to one another with a clear delineation of the phase boundary. This microstructural analysis highlights the robust structural integrity and chemical stability of the Mo_2_TiAlC_2_-based SA.

The X-ray diffraction (XRD, Rigaku, SmartLab SE, Tokyo, HI, Japan) analysis of the Mo_2_TiAlC_2_ powder is illustrated in Figure 4a. The majority of the prominent peaks are identified within the 5°–65° 2*θ* range. The diffraction peaks are observed at 9.46°, 19°, 34.88°, 35.86°, 37.5°, 38.58°, 39.72°, 42.38°, 48.96°, 56.78°, 61.14°, and 61.88°, corresponding to the (002), (004), (011), (012), (013), (008), (014), (015), (017), (019), (0110), and (110) crystal planes of the hexagonal structure of Mo_2_TiAlC_2_, as documented in the existing literature [37]. Notably, the peak at (006) signifies a distinct feature associated with silicon (Si). Since Si powder was not employed as an internal standard in this analysis, it is possible that Si may have been introduced by the crucible during the high-temperature sintering process. The absence of additional impurity phase peaks suggests that the analyzed powder is predominantly a pure phase material. The pronounced sharpness of each peak indicates that the material has undergone complete crystallization, thereby indicating a well-defined layered structure.

The structural characteristics of the Mo_2_TiAlC_2_-based SA were further investigated using a Raman spectrometer (Raman, Horiba, LabRAM HR Evolution, Paris, PMA, France). The vibrational and rotational properties of the molecules from the SA were elucidated by analyzing the frequency variations in the scattered light spectra resulting from excitation with a 532 nm laser. Among the 36 phonon modes of Mo_2_TiAlC_2_, only A_1g_, E_1g_, and E_2g_ exhibited Raman activity [38]. As depicted in Figure 4b, four distinct characteristic peaks are observed within the spectral range of 80–1200 cm^−1^; specifically, the peaks at 208.7 and 704.2 cm^−1^ correspond to the A_1g_ mode associated with the bending of the Mo-C bond. The other two characteristic peaks at 132.6 and 611.2 cm^−1^ closely align with the theoretical values for the E_1g_ mode, and the deviations are attributed to variations in the material composition ratios. The E_1g_ mode involves the out-of-phase movement of C and Mo atoms either in the *x-y* plane or in the *z* direction, while keeping the Al and Ti elements frozen in the structure. These all indicate that the self-prepared Mo_2_TiAlC_2_-based SA exhibits a lower defect density, and a superior physical structure and chemical properties.

The lateral dimensions and film thickness of the Mo_2_TiAlC_2_-based SA were characterized using atomic force microscopy (AFM, Bruker, Dimension Icon, Billerica, MA, USA), which directly measures the height of the sample surface based on the force applied to a microprobe, thereby providing insights into the surface topography. Figure 4c depicts the surface height of the SA at a scale of 2 μm, revealing a film thickness that ranges from 0.75 to 2.1 nm, and indicating that the particle size of Mo_2_TiAlC_2_ is less than 1 μm. The surface roughness appears remarkably uniform, with only a few membranes being slightly thicker, suggesting that the self-prepared SA film is consistent and possesses an appropriate thickness. The red line within the inset represents a measurement of the cross-section height distribution, demonstrating that the average thickness of the SA here is approximately 1.33 nm. After a long time of ultrasonic treatment, the micro nanosheets gathered during the spin coating process, and the local thickened circles were shown [39]. Figure 4d demonstrates the linear absorption of the Mo_2_TiAlC_2_-based SA with the laser wavelength. By inserting a Mo_2_TiAlC_2_-based SA after the output laser, the power before and after inserting the SA were measured. It is observed that when the pump power is increased, a 95.5% linear absorption of the SA is obtained for the output laser.

### 2.3. Experimental Setup of PQS Laser

To characterize the nonlinear optical properties of the Mo_2_TiAlC_2_ MAX phase material serving as an SA, a PQS laser was employed to assess its performance, and the experimental setup of the PQS Tm:YAP laser is depicted in Figure 5. A fiber-coupled laser diode (LD) was used as the pump source, providing a maximum output power of 30 W with a central wavelength of 792 nm at room temperature. The core diameter of the fiber was 105 μm, with a numerical aperture of 0.22. A *b*-axis-cut Tm:YAP crystal, doped with 3 at%. Tm^3+^ concentration, was utilized as the laser gain medium with dimensions of 3 × 3 × 7 mm^3^. Both end faces of the crystal were coated with anti-reflectivity films at 790–810 nm and 1900–2100 nm, then encased in indium foil and placed in a copper heat sink. The heat sink was connected with a water-cooling system in order to efficiently dissipate the heat produced within the crystal. In this experiment, a convenient short resonator cavity was adopted to generate the pulsed-laser output. R1 was a flat mirror coated on one side with reflective film and positioned at a 45° angle, serving to redirect the pump light. The lenses L1 (with a focal length of 25 mm) and L2 (with a focal length of 30 mm) were employed for collimating and focusing the pump beam, respectively. The resonator cavity was composed of two lenses, a flat–concave input mirror M1 and an output coupling mirror (OC), with a total physical length of 23 mm. The curvature radius of M1 was 200 mm, which exhibited high transmittance at 790–810 nm on both faces, and high reflectance at 1900–2100 nm on the concave face. A plane mirror coated with 2% or 5% transmittance (*T*) at the emitted laser wavelength was used as the OC. The Mo_2_TiAlC_2_ film coated on the CaF_2_ substrate, serving as the modulator, was inserted between the Tm:YAP crystal and the OC. R2 was also a 45° angle flat mirror displaying high reflectivity for the emitted laser. The output of the pulsed laser was achieved by adjusting the distance between OC and the SA.

## 3. Results and Discussion

The output performances of the PQS Tm:YAP laser with the Mo_2_TiAlC_2_-based SA are illustrated in Figure 6 and Figure 7. In these experimental evaluations, the output power was quantified using a power meter (PM100D, Thorlabs, Newton, NJ, USA), while a high-speed InGaAs photodetector (ET-5000, EOT, Ann Arbor, MI, USA) was utilized to capture the PQS pulses. This detector was interfaced with a digital oscilloscope (MDO34, Tektronix, Beaverton, OR, USA) featuring a bandwidth of 1 GHz to visualize the pulse waveform. Furthermore, the output laser spectrum was accurately measured using a wavelength meter (772B-MIR, Bristol, Rochester, NY, USA), and both the beam profile and quality of the output laser were assessed through a beam quality analyzer (M2MS-BP209IP2, Thorlabs, Newton, NJ, USA).

Figure 6a demonstrates how the output power of the PQS Tm:YAP laser varied with the incident pump power, in contrast to its behavior in the continuous wave (CW) mode. In the CW mode, it can be observed from the black/red line in the graph that a maximum output power of 1.01/1.16 W was attained with an OC of *T* = 2/5% at 2 μm, corresponding to an optical–optical conversion efficiency of 22/25.3% under a pump power input of 4.59 W. In the PQS mode, as shown by the orange/blue line, an average output power of 273/292 mW was achieved with an OC of *T* = 2/5%, resulting in an optical–optical conversion efficiency of 5.95/6.36% when the Mo_2_TiAlC_2_ MAX phase material was inserted into the resonator cavity as the SA. Due to the low transmittance of the OC, the output power of the CW/PQS Tm:YAP laser was low because of the high losses in the resonator cavity.

Figure 6b illustrates the relationship between the pulse width and pulse repetition frequency (PRF) of the PQS Tm:YAP laser as a function of the absorbed pump power. As the pump power increases from 2.35 to 4.59 W, the pulse width decreases from 858.2/714 to 259.5/242.9 ns, while an upward trend is observed in the PRF, which varies from 32.44/34.78 to 60.25/47.07 kHz for an OC of *T* = 2/5%. Notably, the pulse width of the PQS Tm:YAP laser with an OC of *T* = 5% at the highest pump power is narrower than that with an OC of *T* = 2%, due to a more optimal mode matching between the pump light and oscillation light. To the best of our knowledge, this represents the shortest pulse width, of 242.9 ns, that has been achieved using a PQS solid-state laser in the 2 μm range, with a MAX phase material functioning as the SA.

Figure 6c depicts the typical pulse trains of the PQS Tm:YAP laser at the highest pump power for an OC of *T* = 5% on 2.5, 100 μs/div, and 1ms/div time scales, both exhibiting stable oscillatory behavior. An ultrashort pulse duration of 242.9 ns was recorded at a PRF of 47.07 kHz for the 2.5 μs time scale. Furthermore, a single pulse energy of 6.2 μJ was calculated based on a pulse width of 242.9 ns, a PRF of 47.07 kHz, and an average output power of 292 mW from the PQS Tm:YAP laser. The output spectrum for both the CW and PQS operations is presented in Figure 6d. At an absorbed pump power of 4.59 W, a central output wavelength of 1937.3 nm is achieved in the CW mode, while a central output wavelength of 1931.2 nm is recorded during the PQS operation. It can be observed that the central output wavelength of the PQS-mode laser is less than that of the CW mode. The blue shift can be attributed to the lower energy storage in the stimulated-emission cross section of the Tm:YAP crystal in the CW mode than PQS operation [40]. Additionally, the insertion loss induced by the Mo_2_TiAlC_2_-SA further contributed to the shorter output wavelength obtained from the PQS Tm:YAP laser [41].

The beam quality of the PQS Tm:YAP laser at a pump power of 4.59 W for a *T* = 5% OC is illustrated in Figure 7a. A lens with a focal length of 200 mm was employed to focus the emission laser onto the beam quality analyzer. The beam quality factors were measured as *M_x_*^2^ = 1.11 and *M_y_*^2^ = 1.27 at the maximum average output power, corresponding to an initial divergence angle of 0.894° and 1.016°, respectively. Furthermore, both the 2D and three-dimensional (3D) images of the laser profile are presented in the inset pictures, indicating that the beam shape of the PQS Tm:YAP laser approached a diffraction-limited performance. As depicted in Figure 7b, the output power stability of the PQS Tm:YAP laser was assessed by calculating the ratio of the root mean square (RMS) of the collected data to the average output power. A minimal deviation of only 0.34% was recorded every half-second over a three-hour period with an average output power of 292 mW, corresponding to an OC of *T* = 5%. This indicates that the output power of the PQS Tm:YAP laser utilizing Mo_2_TiAlC_2_-SA as the modulator exhibited enhanced stability.

Table 1 summarizes the output performances of the PQS solid-state laser for a 2 μm wavelength range with MoS_2_, WSe_2_, WSe_2_/CuO heterojunction, SnS_2_, TaTe_2_, Nb_2_CT_x_, V_2_CT_x_, Ta_4_AlC_3_, and Mo_2_TiAlC_2_ SAs. From Table 1, it can be seen that the stable ultrashort pulse generated in this work has a duration of 242.9 ns, representing the narrowest pulse width obtained for PQS solid-state lasers utilizing various 2D materials as the SAs. The resonator cavity we designed has a physic length of merely 23 mm; however, the single pulse energy and PRF remain competitive due to the exceptional nonlinear optical properties exhibited by the Mo_2_TiAlC_2_-based SA. Among the MAX phases-based SAs, the PQS solid-state laser with a Mo_2_TiAlC_2_-based SA has excellent performance across nearly all aspects.

## 4. Conclusions

In conclusion, we presented a convenient spin-coating method for utilizing a Mo_2_TiAlC_2_ MAX phase ceramic material to prepare an SA for a PQS Tm:YAP laser. The home-made Mo_2_TiAlC_2_ film exhibited a well-defined lamellar morphology, robust structural integrity, uniform thickness, and an effective strong interfacial bonding with the substrate. We studied a PQS Tm:YAP laser by employing a Mo_2_TiAlC_2_-based SA as the modulator. With a compact resonant cavity of 23 mm, a 242.9 ns stable ultrashort pulse train was acquired at 47.07 kHz, representing the narrowest pulse width reported for PQS solid-state lasers utilizing MAX phase materials as SAs. Meanwhile, the average output power reached 292 mW with a central output wavelength of 1931.2 nm, while the absorbed pump power was 4.59 W. The beam quality factors were measured to be *M_x_*^2^ = 1.11 and *M_y_*^2^ = 1.27, respectively, and the PQS Tm:YAP laser exhibited remarkable stability. To the best of our knowledge, this is the first report on a PQS solid-state laser utilizing a Mo_2_TiAlC_2_ MAX phase material as an SA, thereby paving the way for its broader applications in mid-infrared solid-state lasers.

## Figures and Tables

**Figure 1 nanomaterials-14-01823-f001:**
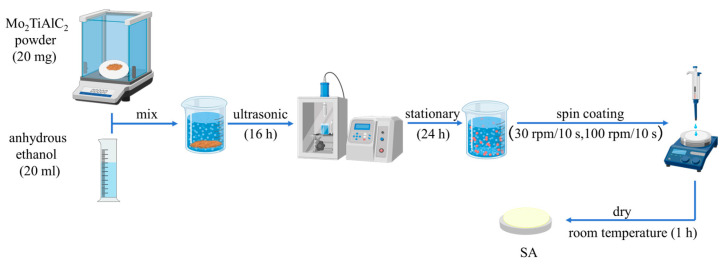
Illustration of the preparation process for the Mo_2_TiAlC_2_-based SA.

**Figure 2 nanomaterials-14-01823-f002:**
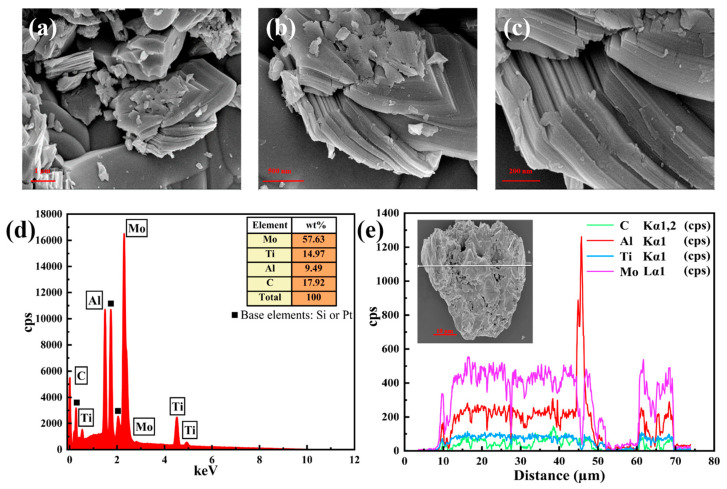
Morphological characterization of the Mo_2_TiAlC_2_ powder. (**a**–**c**) SEM images of the Mo_2_TiAlC_2_ powder at different magnifications; (**d**,**e**) EDS images of the Mo_2_TiAlC_2_ powder.

**Figure 3 nanomaterials-14-01823-f003:**
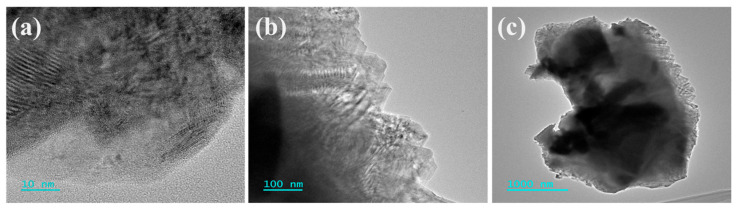
(**a**–**c**) TEM images of the Mo_2_TiAlC_2_ powder at different magnifications.

**Figure 4 nanomaterials-14-01823-f004:**
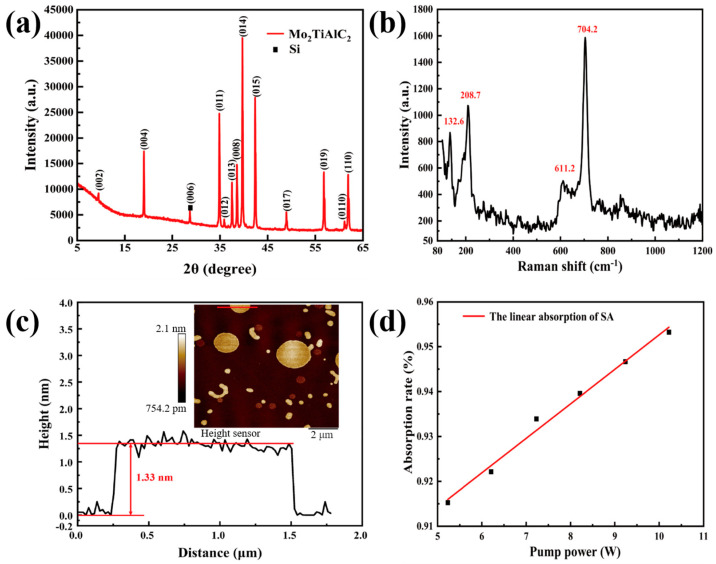
Structural and linear absorption characterization of Mo_2_TiAlC_2_ material. (**a**) XRD image of Mo_2_TiAlC_2_ powder; (**b**) Raman spectrum of Mo_2_TiAlC_2_ powder; (**c**) AFM image of Mo_2_TiAlC_2_-based SA; and (**d**) linear absorption properties of Mo_2_TiAlC_2_-based SA.

**Figure 5 nanomaterials-14-01823-f005:**
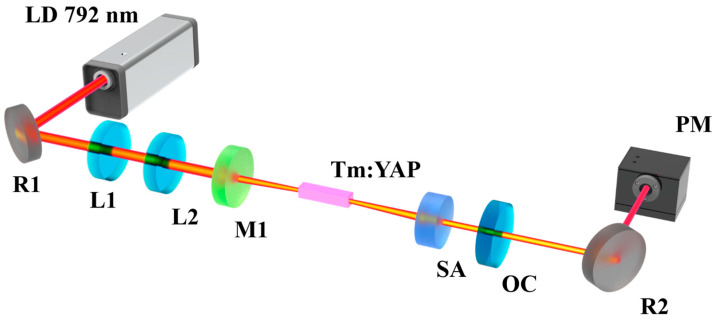
The experimental setup of the PQS Tm:YAP laser with the Mo_2_TiAlC_2_-based SA.

**Figure 6 nanomaterials-14-01823-f006:**
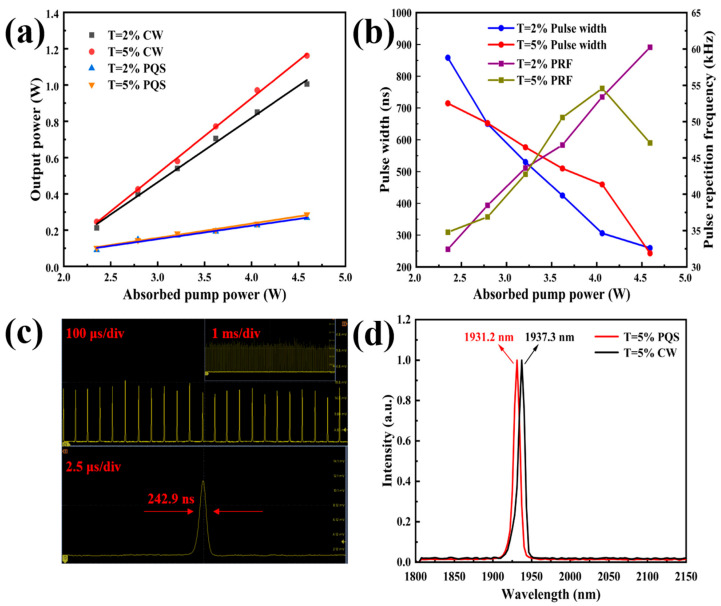
Output performances of PQS Tm:YAP laser. (**a**) Output power varies with absorbed pump power; (**b**) pulse width and repetition frequency as a function of absorbed pump power; (**c**) stable pulse trains of *T* = 5% (2.5, 100 μs/div, and 1 ms/div); and (**d**) output spectrum of laser in CW and PQS modes.

**Figure 7 nanomaterials-14-01823-f007:**
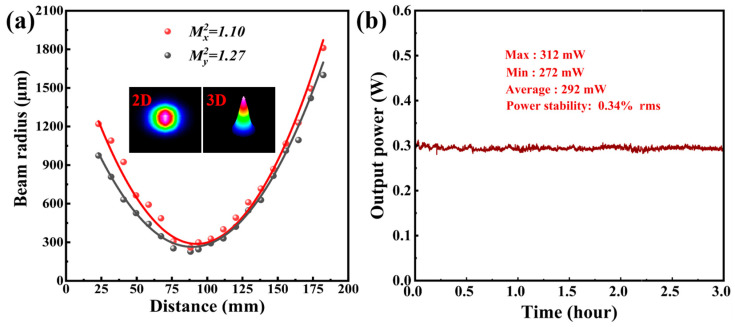
(**a**) The beam quality of the PQS Tm:YAP laser with *T* = 5%; (**b**) output power stability of the PQS Tm:YAP laser.

**Table 1 nanomaterials-14-01823-t001:** A comparison of the output performances of PQS solid-state lasers with different SAs.

SAs	Crystals	Wavelength (nm)	Pulse Width (ns)	Repetition Frequency (kHz)	Pulse Energy (μJ)	Output Power (mW)	Year	Ref.
MoS_2_	Tm:YAP	1936.0	2500	24.0	3.8	100	2020	[42]
WSe_2_	Tm:YAP	1988.3	392	113.7	11.4	1290	2020	[43]
WSe_2_/CuO	Tm:YAP	1986.8	752	68.68	33.7	2320	2024	[44]
SnS_2_	Tm:YAP	1987.9	3061	70	12.4	621	2023	[45]
TaTe_2_	Tm:YAP	1987.9	585	191.1	9.7	1860	2023	[46]
Nb_2_CT_x_	Tm:YAP	1938.0	1960	80.0	7.7	620	2021	[47]
V_2_CT_x_	Tm:YAP	1937.4	528	65.9	6.3	350	2023	[48]
Ta_4_AlC_3_	Tm:YAP	1991.8	926	143.8	5.4	780	2023	[49]
Mo_2_TiAlC_2_	Tm:YAP	1931.2	242.9	47.07	6.2	292	2024	This work

## Data Availability

The data that support the findings of this study are available from the corresponding authors upon reasonable request.

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
