# Peer review of "Mo_2_TiAlC_2_ as a Saturable Absorber for a Passively Q-Switched Tm:YAlO_3_ Laser"

_nanomaterials, 2024, doi:10.3390/nano14221823_

Round 1
Reviewer 1 Report
Comments and Suggestions for Authors
Review on the paper entitled: “Mo2TiAlC2 as a Saturable Absorber for a Passively Q-switched Tm:YAlO3 laser”. The paper is well organized and written. Before a possible publication, it needs some minor revisions. I’m not an expert of the optical characterization of the laser properties, and for this reason my comments and suggestions reported below are focused on the structural characterizations:
-After the spin-coating and drying process, is a film obtained on the saturable adsorber?? What is the typical thickness and roughness of these Mo2TiAlC2 films?
-Do these nanosheets maintain van der Waals distance between the different layers comparable with those of classic Transition Metal Dichalcogenides?
-For this specific laser applications, have these nanosheets agglomerates some advantages in terms of performance compared to homogeneous films?
-After the Raman evaluation, what do the authors mean for superior physical structures and chemical properties on these materials? It is not perfectly clear.
-Is it possible to evaluate strain/doping by Raman measurements for these vdW systems?
-AFM characterization is quite different from those obtained by SEM images..I understand that is difficult to have the same x-y resolution, but can the authors carry out some better evaluation of these nanosheets? These circular features are quite strange..
Reviewer 2 Report
Comments and Suggestions for Authors
Dear Authors,
Your paper 'Mo2TiAlC2 as a Saturable Absorber for a Passively Q-switched Tm:YAlO3 laser' is dealing with the production, characterization and application of a novel saturable absorber material. I understand that you tried to cover all the interests from the material supplier to a laser developer, that's why the paper is finally a little long. The problem is that from the standpoint of a laser developer I completely missing the main parameters like saturation energy density, recovery time, unsaturable losses etc of the material. There is not even a simple absorption spectrum shown. Without these information I can guess I would be able to rebuild the laser you have described but I am not able to develop my own. In particular the unsaturable losses seem to be rather high, what does not only result in a lower laser efficiency but may limit the total power a laser comprising that saturable absorber. From the standpoint of a material developer I would not be interested in the laser details. My suggestion would be to split the paper into two, but this is up to you to keep it that way.
In detail I have the following issues with the paper:
1) In Figure 6c you show a pulse train of about 20 pulses, where the difference between the first and last pulse is larger than 20%. Moreover there is a obvious slope in the pulse energy. It would be interesting to see a zoomed out view of the measured trace, what could be added to convince the reader that there is no additional oscillation on the output.
2) On page 6 you state: 'Due to the low transmittance of OC, the output power of the CW/PQS Tm:YAP laser was low because of the high losses in the resonator cavity.' This is not clear to me. Would that mean that by adapting the OC a higher power could be generated? Obviously the high loss is coming from your saturable absorber. Are these losses diffractive or absorptive in nature?
3) What was the linear absorption of the SA at the laser wavelength?
4) On page 6/7 you explain: the pulse width in PQS Tm:YAP laser with an OC of T = 5% at the highest pump power was narrower than that with an OC of T = 2%, due to a more optimal mode matching between the pump light and oscillation light. I do not understand how the different Transmission of the OC change the resonator mode, what should be mainly given by the length of the cavity and mirror curvature. By adjusting the cavity length you should be able to tune it to the optimum. Was this done or if not why?
5) In figure 6d you show the difference in output wavelength between cw and pulsed mode. The question here is if the cw laser spectrum was recorded at the same output power as with q-switch? You should better explain what is meant by 'The longer output wavelength noted in CW mode compared to that in PQS operation can be attributed to a lower energy storage within the stimulated emission cross-section of the Tm:YAP crystal under CW condition than that during PQS operation.' Why is energy storage dependent on pulsed or cw mode? The fluorescence life time of Tm:YAP is much longer than the time between your laser pulses, so the laser crystal effectively averages the power and no wavelength shift should be observable. The shift in wavelength could have a couple of reasons for instance a wavelength dependent behavior of the absorber or an inversion dependent gain in Tm:YAP. How you can exclude some?
In conclusion I have voted for 'accept after minor revision' if you really want to keep the paper closely to its actual content.
Round 2
Reviewer 1 Report
Comments and Suggestions for Authors
I'm sorry for the reply. But the authors do not answered to me in questions n.5 and 6.
In particular, n.5 I do not asked to them to explain me how to evaluate strain and doping effects from Raman. I perfectly know the correlation between vibrational characteristics and strain/doping effects induced on the materials. I asked to the authors if they can evaluate strain and doping effects by Raman spectroscopy in "their 2D materials systems". A negative response is also possible.
AFM interpretations and explainations are not clear in comparison to the SEM images.
Reviewer 2 Report
Comments and Suggestions for Authors
Dear Authors,
Thank's for considering these suggestions. I agree to publish the paper as it is.
Author Response
Comments 1: [Dear Authors,Thank's for considering these suggestions. I agree to publish the paper as it is.]
Response 1:[Thank you for you comments and suggestions.]
Round 3
Reviewer 1 Report
Comments and Suggestions for Authors
No further comments.